# *Hand2os1* Regulates the Secretion of Progesterone in Mice Corpus Luteum

**DOI:** 10.3390/vetsci9080404

**Published:** 2022-08-02

**Authors:** Yanni Jia, Lu Liu, Suhua Gong, Haijing Li, Xinyan Zhang, Ruixue Zhang, Aihua Wang, Yaping Jin, Pengfei Lin

**Affiliations:** 1College of Veterinary Medicine, Northwest A&F University, Xianyang 712100, China; ynj@nwafu.edu.cn (Y.J.); liulucvm@nwafu.edu.cn (L.L.); gongsuhua@nwafu.edu.cn (S.G.); lihaijing@nwafu.edu.cn (H.L.); xinyanzhang@nwafu.edu.cn (X.Z.); zhangruixue@nwafu.edu.cn (R.Z.); wangaihau@nwafu.edu.cn (A.W.); 2Key Laboratory of Animal Biotechnology, Ministry of Agriculture and Rural Affairs, Northwest A&F University, Xianyang 712100, China

**Keywords:** lncRNA, *Hand2os1*, mouse, progesterone, corpus luteum

## Abstract

**Simple Summary:**

The corpus luteum plays a key role in pregnancy maintenance and estrous cycle regulation by secreting progesterone. In this study, we investigate the expression and regulation of lncRNA *Hand2os1* in the ovaries. We found *Hand2os1* was specifically detected in luteal cells during the proestrus and estrus phases, and strongly expressed in the corpus luteum on day 4 and day 18 of pregnancy. Moreover, *Hand2os1* regulates the secretion of progesterone in the mouse corpus luteum by affecting the key rate-limiting enzyme StAR, which suggests it may have an impact on the maintenance of pregnancy.

**Abstract:**

The corpus luteum plays a key role in pregnancy maintenance and estrous cycle regulation by secreting progesterone. *Hand2os1* is an lncRNA located upstream of *Hand2*, with which a bidirectional promoter is shared and is involved in the regulation of cardiac development and embryo implantation in mice. The aim of this study was to investigate the expression and regulation of *Hand2os1* in the ovaries. Here, we used RNAscope to detect differential expression of *Hand2os1* in the ovaries of cycling and pregnant mice. *Hand2os1* was specifically detected in luteal cells during the proestrus and estrus phases, showing its highest expression in the corpus luteum at estrus. Additionally, *Hand2os1* was strongly expressed in the corpus luteum on day 4 of pregnancy, but the positive signal progressively disappeared after day 8, was detected again on day 18, and gradually decreased after delivery. *Hand2os1* significantly promoted the synthesis of progesterone and the expression of *StAR* and *Cyp11a1*. The decreased progesterone levels caused by *Hand2os1* interference were rescued by the overexpression of *StAR*. Our findings suggest that Hand2os1 may regulate the secretion of progesterone in the mouse corpus luteum by affecting the key rate-limiting enzyme StAR, which may have an impact on the maintenance of pregnancy.

## 1. Introduction

The corpus luteum (CL) is a gland formed after ovulation with a temporary endocrine function. The main role of the CL is to secrete progesterone to maintain pregnancy. The CL can be divided into cyclical and pregnancy CLs. If pregnancy does not occur, the CL will degenerate into a white body under the action of prostaglandin F2α (PGF_2α_) and prolactin (PRL), and then the female animal will enter the next round of the estrus cycle [1,2,3]. Progesterone (P_4_), secreted by the CL, plays a crucial role in regulating the estrous cycle and maintaining pregnancy. Luteal cells utilize steroidogenic acute regulatory protein (StAR), P450 cholesterol side chain cleavage enzyme (P450scc; encoded by CYP11A1), and 3β-hydroxysteroid dehydrogenase (3β-HSD) to produce progesterone, and this process is regulated tightly. [4]. Ovulation occurs with a sudden increase in luteinizing hormone (LH) levels, followed by luteinization of the granulosa cells (GC), where StAR expression increases. P450scc then catalyzes the conversion of cholesterol to pregnenolone, which is further metabolized to progesterone via 3β-HSD. [4,5,6].

It has been confirmed that lncRNAs can participate in many aspects of gene function and regulation; they mainly interact with mRNAs, DNA, proteins, and miRNAs, and regulate genes in multiple ways [7,8]. At present, the roles of lncRNAs in the field of reproduction have been widely reported, including spermatogenesis, oocyte and embryo development, follicular development and ovulation, and placenta formation [9,10,11]. Additionally, a study found that the lncRNA *Neat1* was highly expressed in the CL, and that severe damage to the CL was observed in nearly half of *Neat1* knockout mice studied. These data suggest that *Neat1* may play a role in maintaining ovarian function [12]. Moreover, it was confirmed that StAR and progesterone production is disturbed in the H19KO mouse model [13]. However, the expression and biological function of lncRNAs in the CL remain unclear.

*Hand2os1* (*Uph*, also known as *HAND2-AS1* in humans) is located upstream of *Hand2* and shares a bidirectional promoter with it [14]. *Hand2os1* can inhibit the proliferation of cardiomyocytes by inhibiting the expression of *Hand2*, thereby reconciling the balance of cardiomyocyte lineages [15]. Previous studies have shown that the lncRNA *Hand2os1* is involved in the regulation of decidualization of the mouse uteri and that it is regulated by P_4_ [16]. To delineate the role of *Hand2os1* in the CL, we collected ovaries at different stages, constructed in vitro and in vivo models of CL formation and degeneration, and used qRT-PCR and RNAscope to explore further biological functions. Taken together, our findings elucidate the physiological functions of *Hand2os1* in the production of P_4_ and provide evidence for antisense lncRNAs in regulating luteal function.

## 2. Materials and Methods

### 2.1. Ethics Statement

The methods used in this study were performed in accordance with the guidelines of the Committee for Ethics on Animal Care and Experiments at Northwest A&F University. All experimental protocols involving animal subjects had received prior approval from the Experimental Animal Manage Committee, and the approval license number was 2017ZX08008005.

### 2.2. Animals, Treatments, and Sample Collection

Adult mice (Kunming strain), 8–10 weeks old, were purchased from the Chengdu Dashuo Experimental Animal Center (Chengdu, China). Animals were housed under a 12 L:12 D cycle and provided with food and water ad libitum.

Sexually mature mice were selected, and the vaginal smear method and hematoxylin and eosin (HE) staining were used to further identify the estrus cycle phase. Mice with different estrus cycle phases (*n* = 3 for each phase) were selected and sacrificed using the dislocation method, and ovarian tissue from both sides was collected.

To establish a model of CL formation and degeneration, immature female mice were divided into two groups at 21 days of age. The experimental group (*n* = 3) was intraperitoneally injected with PMSG (5 IU) and, 48 h later, with hCG (5 IU); the control group (*n* = 3) was injected with saline [12]. Mice were sacrificed by cervical dislocation at 0, 24, 48, 72, and 96 h after the second injection, and the ovarian tissues from both sides of the body were collected.

Sexually mature female mice were mated with fertile male mice of the same breed to induce pregnancy. Pregnancy was confirmed on days 1 and 4 (day 1 = vaginal congestion day) by recovering embryos from the fallopian tubes and uterus, respectively. Mice were killed at 9 a.m. by cervical dislocation on pregnancy days 0 (D0), D1, D2, D4, D8, D11, D14, and D18 as well as at postpartum days 1 (PD1), PD3, and PD5 (*n* = 3 each). Ovarian tissue samples were collected from both sides of the body.

### 2.3. Detecting Target RNA via RNAscope

Ovarian tissues were formalin-fixed, paraffin-embedded, and sliced into 5 μm-thick sections. Target gene expression was detected using a *Hand2os1*-specific targeting probe (Advanced Cell Diagnostics, Silicon Valley, CA, USA). Follow the instructions for the RNAscope^®^ 2.5 HD detection kit (Advanced Cell Diagnostics). Finally, the tissue sections were counterstained with hematoxylin. Images were acquired using a fluorescence Ni-U microscope (Nikon, Tokyo, Japan).

### 2.4. Isolation and Culture of Mouse Luteum Cells

Four-day pregnant mice were sacrificed by cervical dislocation at 9 a.m., and bilateral ovarian tissues were collected under aseptic conditions. After washing with PBS 2–3 times, ophthalmic forceps were used to peel off the CL tissue inlaid in the ovary under a stereoscope. The CL tissue was incubated with 0.1% collagenase II (Sigma, St. Louis, MO, USA) for 40 min at 37 °C, and digestion was stopped with GibcoDulbecco’s Modified Eagle Medium: F-12 (DMEM/F12) medium containing 10% serum. Digested tissues were filtered through a 70 µm filter (Cell Strainer; Millipore, MA, USA) and precipitated cells were collected [17]. All the cultured cells were maintained in a humidified incubator at 37 °C with 5% CO_2_. The luteal degeneration model was constructed using mouse primary luteal cells cultured in medium containing PGF_2α_ (1 µM) for 24 h [18].

### 2.5. Isolation, Culture, and Luteinization of Mouse Granulosa Cells

Mice in the estrus phase that had been detected by HE staining were killed by cervical dislocation, and bilateral ovarian tissues were collected under aseptic conditions. In DMEM/F-12 medium, a needle was used to pierce the follicle on the ovary to release granulosa cells. The collected solution was filtered through a 70-µm filter (Cell Strainer; Millipore, MA, USA). The cells were then seeded in DMEM/F12 culture medium with 100 U/mL penicillin, 100 μg/mL streptomycin, and 10% charcoal-treated fetal bovine serum (FBS; Life Technologies, Carlsbad, CA, USA) at a concentration of 2 × 10^5^ cells/well in 35-mm dishes. The granulosa cells were treated with 100 ng/mL luteinizing hormone (LH) to induce luteinization and harvested after 0, 12, and 24 h [19].

### 2.6. Transfection of Hand2os1 siRNA and Overexpression Vector

Mouse luteal cells (LCs) were transfected either with a *Hand2os1* siRNA or negative control, designed, and synthesized by Gene Pharma Co., Ltd. (Shanghai, China), using TurboFect Transfection Reagent (Thermo Scientific, Shanghai, China), according to the manufacturer’s instructions. Overexpression experiments were performed using the previously constructed vector pcDNA3.1-*Hand2os1* [16].

### 2.7. Cell Proliferation Assay

LCs were plated in 96-well plates at a density of 1 × 10^4^ cells/well and treated with Hand2os1 siRNA and overexpression vector after 24 h of culture. Then, the instructions of the Cell Counting Kit-8 (Beyotime, Shanghai, China) were followed to conduct the experiment, using a microplate reader 680 (Bio-Rad Laboratories Inc., Hercules, CA, USA) to detect the absorbance. All experiments were performed in triplicate.

### 2.8. P_4_ Level Detection by ELISA

According to the experimental requirements, the culture supernatants of LCs and GCs were collected, pretreated by centrifugation at 3000× *g* at 4 °C for 20 min, and then transferred to a clean centrifuge tube. Conditioned media were collected for the analysis of P_4_ content using a P_4_ ELISA kit according to the manufacturer’s instructions (Mlbio, Shanghai, China). Absorbance at 450 nm was measured using a microplate reader 680 (Bio-Rad).

### 2.9. Cell Apoptosis Assay

LCs were plated into 6-well plates and incubated for 24 h. Apoptosis was assessed using an Annexin V-FITC/PI Apoptosis Detection Kit (Keygen Biotech, Nanjing, China). The samples were subjected to flow cytometry using a FACSAria III Cell Sorter (BD Biosciences, San Jose, CA, USA). All data were analyzed using FlowJo X 10.0.7 software (Palo Alto, CA, USA).

### 2.10. qRT-PCR Analysis

Total RNA was extracted using TRIzol reagent (Invitrogen, Shanghai, China). RT was performed using the 5× All-In-One RT Master Mix with the AccuRT Genomic DNA Removal Kit (Applied Biological Materials, Inc.; Vancouver, BC, Canada). qRT-PCR was performed using the Eva Green qPCR Master Mix Kit (Vazyme Biotech Co., Ltd., Nanjing, China) on a CFX96 Real-Time PCR Detection System (Bio-Rad Laboratories Inc.). All reactions were conducted in biological triplicates. The primers and temperatures used are listed in Table 1.

### 2.11. Data Analysis and Statistics

Each experiment was repeated at least 3 times in each group. GraphPad Prism 6 software (GraphPad Software, San Diego, CA, USA)was used for statistical analysis of data, and one-way ANOVA or independent samples *t*-test was used for significant difference analysis. *p* < 0.05 means significant difference.

## 3. Results

### 3.1. Differential Expression of Hand2os1 during the CL Development

To evaluate the physiological functions of *Hand2os1* in the CL, RNAscope was first performed to analyze the spatial distribution of this lncRNA in the ovary during the estrus cycle and pregnancy (Figure 1). In all the ovaries studied, *Hand2os1* was specifically expressed in LCs; no positive staining for *Hand2os1* was observed in any other cell type. During the estrous cycle, highly positive staining for *Hand2os1* was detected in the cytoplasm of LCs during the estrus phase compared to that found during the proestrus phase (Figure 1A). In contrast, no positive staining for *Hand2os1* was observed in the CL during the diestrus and metestrus phases. During pregnancy, no positive signal for *Hand2os1* was detected in the CL on D1, but *Hand2os1* expression was strongly detected in the cytoplasm of CL cells on D4 and then disappeared from D8 to D14 (Figure 1B). Notably, a strong *Hand2os1* signal peaked again in the nucleus of the LCs during the delivery period (D18). Then, the expression of *Hand2os1* gradually decreased from PD1 to PD3. These results indicated that *Hand2os1* might be involved in CL formation and regression. Surprisingly, no positive signal for *Hand2os1* was detected in the hCG-induced CL formation and regression model (Figure 2).

### 3.2. Effects of Hand2os1 on LC Proliferation and Apoptosis

The proliferation of LCs isolated from the CL on D4 of pregnancy was not affected by silencing or overexpression of *Hand2os1* for 24 h (Figure 3A,B). To further analyze the effect of *Hand2os1* during LC degeneration, a model of luteal regression was established by the addition of PGF_2α_ (1 µM). Flow cytometry results showed that overexpression of the *Hand2os1* gene did not change PGF_2α_-induced LCs apoptosis (Figure 3C).

### 3.3. The Effect of Hand2os1 on the Formation of the CL

Follicular granulosa cells differentiate into LCs in response to LH stimulation. In this study, primary granulosa cells were cultured and luteinized by a 100 ng/mL LH treatment. As shown in Figure 4, the genes encoding the key steroidogenic enzymes *StAR* and *Cyp11a1* were significantly upregulated with increasing LH treatment time (*p* < 0.05); however, the expression level of *Hand2os1* mRNA was not significantly altered by LH treatment. Furthermore, *StAR* and *Cyp11a1* mRNA levels were unaffected by *Hand2os1* overexpression in LH-treated luteinized granulosa cells (Figure 5). ELISA results further confirmed that *Hand2os1* overexpression had no effect on progesterone secretion during granulosa cell luteinization (Figure 5).

### 3.4. Effect of Hand2os1 on Progesterone Synthesis in LCs

To further explore the role of *Hand2os1* in progesterone synthesis, we either silenced or overexpressed the *Hand2os1* gene in cultured primary LCs isolated from the CL of D4 of pregnancy samples. The results showed that silencing of *Hand2os1* significantly suppressed progesterone production in cultured primary LCs compared to that in control cells (Figure 6A, *p* < 0.05). Conversely, *Hand2os1* overexpression markedly promoted progesterone production in cultured primary LCs (Figure 6A, *p* < 0.05). In addition, the mRNA levels of the steroidogenic enzymes *StAR* and *Cyp11a1* were significantly decreased or increased after the silencing or overexpression, respectively, of the *Hand2os1* gene in cultured primary LCs (Figure 6B,C, *p* < 0.05).

To verify the mechanism underlying *Hand2os1* regulation of progesterone synthesis through steroidogenic enzymes, we constructed a *StAR* overexpression vector and used it to transfect primary LCs from *Hand2os1*-silenced mice. As shown in Figure 7A,B, *StAR* overexpression did not change *Hand2os1* expression in mouse primary LCs, but significantly upregulated the mRNA expression of *StAR* inhibited by *Hand2os1*. Importantly, overexpression of *StAR* significantly increased the progesterone production that was inhibited by si-*Hand2os1* in mouse primary LCs (Figure 7C, *p* < 0.05).

## 4. Discussion

After ovulation, somatic cells gradually differentiate into LCs. Then, the CL develops into an operational endocrine gland involved in the growth and development of steroid cells [20]. The main function of the CL is to secrete progesterone, which is required to preserve pregnancy in most mammalian species. As important regulatory factors, lncRNAs are involved in the early embryonic development process and in maintaining reproductive ability, including oocyte maturation, zygotic genome activation, and mitochondrial function [21,22,23,24]. However, there are few reports on the effect of lncRNA on the maintenance of the CL in the ovaries during early pregnancy. Our previous study showed that the lncRNA *Hand2os1* is involved in mouse embryo implantation [16]. In this study, we found that *Hand2os1* was highly expressed in the luteal cells of mice during embryo implantation and delivery. This is consistent with the results of a previous study, in which some lncRNAs were found to play a role in embryo implantation and labor through immune responses [25]. These results suggest that *Hand2os1* may be involved in embryo attachment and delivery mechanisms. In addition, highly positive staining for *Hand2os1* was detected in the CL during the estrous phase. However, through hormone-induced CL formation and degeneration, no positive staining of *Hand2os1* was found in immature mouse ovaries, indicating that *Hand2os1* exists only in sexually mature ovaries.

After pregnancy occurs, the CL secretes hormones such as P_4_ to participate in the regulation of implantation and early maintenance of pregnancy [26]. If pregnancy fails or if the fetus is delivered, the CL regresses normally to start a new reproductive cycle [27]. Therefore, the maintenance and regression of the CL are crucial for pregnancy and the continuity of the sexual cycle in females. The formation and degeneration of the CL involve the proliferation and apoptosis of LCs. PGF_2α_ can reduce the concentration of P_4_ in the serum and CL. PGF_2α_ may also participate in the induction of CL cell membrane damage and CL cell apoptosis, leading to the degradation of the CL structure, which involves a variety of cytokines and immune functions [28]. Previous studies have shown that lncRNA *SRA* stimulates mouse granulosa cell growth, changes the cell cycle distribution with an increase in cyclins D1, E, and B, and inhibits cell apoptosis by upregulating *Bcl2* and downregulating *Bax* [29,30]. However, our results showed that neither overexpression nor silencing of *Hand2os1* affected the proliferation and apoptosis of LCs. This suggests that *Hand2os1* has no significant effect on CL formation and degeneration.

P_4_ is a key reproductive hormone in the establishment and maintenance of early pregnancy, and its synthesis is regulated by many factors [4]. Some lncRNAs, such as *SRA*, *H19*, and *Neat1*, also play a role in the production of steroid hormones and the expression of key enzymes [12,31,32]. The main function of the CL is to maintain pregnancy by secreting P_4_. In this study, we found that *Hand2os1* affects the level of P_4_ secreted by LCs during early pregnancy. This result suggests that *Hand2os1* may be involved in the secretion of P_4_ in LCs. StAR and Cyp11a1 are key enzymes in progesterone synthesis [4]. We inhibited *Hand2os1* with a specific siRNA and reduced the expression of the key steroid genes *StAR* and *Cyp11a1*. In contrast, *Hand2os1* overexpression enhanced the expression of *StAR* and *Cyp11a1*. In addition, overexpression of *StAR* rescued the decrease in *StAR* and P_4_ caused by *Hand2os1* interference, further showing the effect of *Hand2os1* on P_4_ synthesis and suggesting that it may affect the secretion of P_4_ through the StAR pathway. Interestingly, *Hand2os1* does not participate in the regulation of key steroid genes (i.e., *StAR* and *Cyp11a1*) during granulosa cell luteinization, implying that its role in different stages of the reproductive process may be quite different. *Hand2os1* is mainly involved in the regulation of progesterone secretion in CL during estrus and pregnancy.

## 5. Conclusions

This study characterized the expression of *Hand2os1* in different luteal phases. Our results show that *Hand2os1* can participate in the function of progesterone secretion in mouse CL by affecting the expression of the key rate-limiting enzyme StAR. Accordingly, this study provides a valuable resource for identifying functional lncRNAs associated with the CL and pregnancy. However, the specific molecular mechanism by which Hand2os1 is involved in the regulation of luteal function requires further study.

## Figures and Tables

**Figure 1 vetsci-09-00404-f001:**
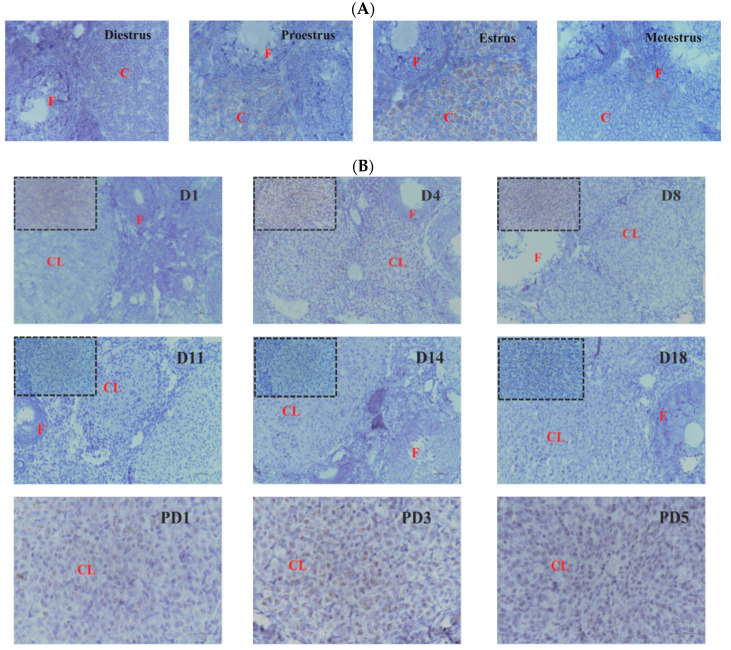
RNAscope of *Hand2os1* expression in the ovary on days 1–18 of pregnancy (D1–D18) and days 1–5 of postpartum (PD1–PD5). Positive expression results in a brown color. (**A**) *Hand2os1* expression and localization in ovaries during the estrus cycle. (**B**) *Hand2os1* expression in ovaries during pregnancy. CL, corpus luteum; F, follicle. Scale bar = 50 µm.

**Figure 2 vetsci-09-00404-f002:**
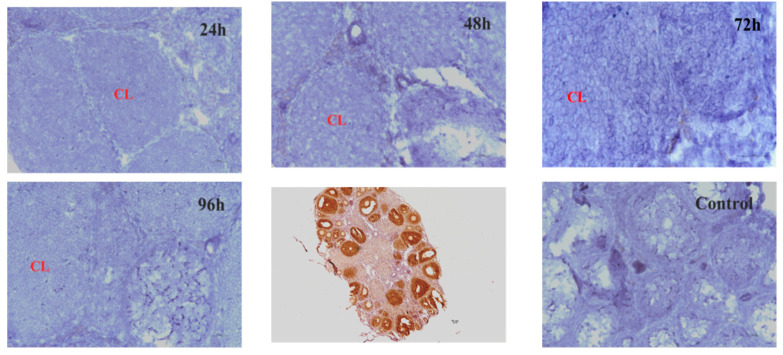
*Hand2os1* expression in the hCG-induced CL formation and regression model for different hours (24 h, 48 h, 72 h, 96 h) was detected by RNAscope. Positive expression results in a brown colour. CL, corpus luteum. Scale bar = 50 µm.

**Figure 3 vetsci-09-00404-f003:**
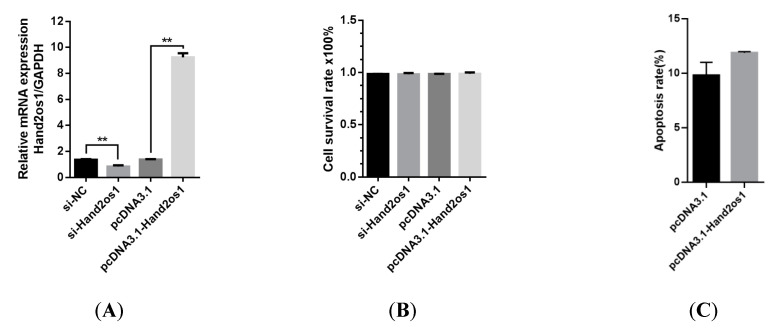
Effects of *Hand2os1* on proliferation and apoptosis in mouse LCs collected on day 4 of pregnancy. (**A**) The efficiency of *Hand2os1* silencing and overexpression were determined using qRT-PCR 24 h after transfection with si-*Hand2os1* and pcDNA3.1-*Hand2os1*, respectively. *GAPDH* was used as the reference gene for normalization. (**B**) The proliferation of LCs was determined by CCK-8 assays after silencing or overexpression of *Hand2os1* for 24 h. (**C**) The apoptosis of LCs was analyzed by FCM after transfection with the *Hand2os1* overexpression vector, followed by treatment with 1 µM PGF_2α_ for 24 h. The data represent the mean ± SEM from three independent experiments. ** *p* < 0.01 compared with si-NC group, and different number of asterisks on bars indicate significant differences (*p* < 0.05).

**Figure 4 vetsci-09-00404-f004:**
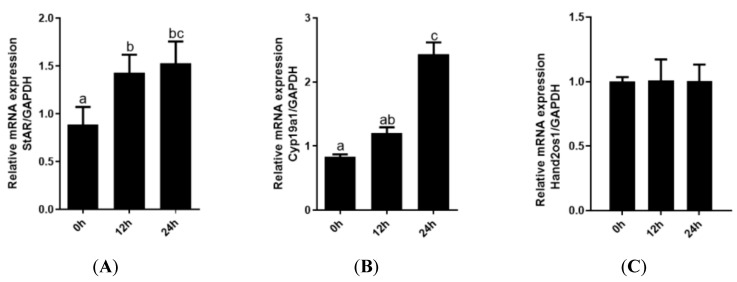
Expression analyses of *StAR*, *Cyp11a1*, and *Hand2os1* during luteinization of granulosa cells. The mRNA levels of (**A**) *StAR*, (**B**) *Cyp11a1*, and (**C**) *Hand2os1* were detected by qRT-PCR after primary granulosa cells were treated with 100 ng/mL LH for 0, 12, and 24 h. *GAPDH* was used as the reference gene for normalization. The data represent the mean ± SEM from three independent experiments. b (*p* < 0.05), c (*p* < 0.01) compared with 0h group (a), the same letter indicates no significant difference and different letters means significant difference (*p* < 0.05).

**Figure 5 vetsci-09-00404-f005:**
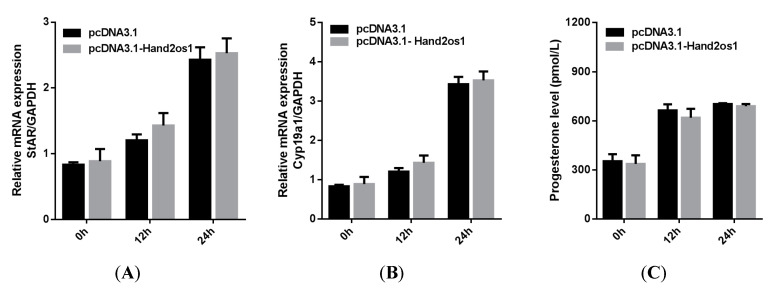
Effects of *Hand2os1* on the expression of steroidogenic enzymes and progesterone production in LH-induced luteinized granulosa cells. The mRNA levels of (**A**) *StAR* and (**B**) *Cyp11a1* were detected by qRT-PCR in 1 IU/mL LH-treated granulosa cells after transfection with pcDNA3.1-*Hand2os1* for 0, 12, and 24 h. (**C**) Meanwhile, the concentration of progesterone in the culture supernatants were determined by ELISA. *GAPDH* was used as the reference gene for normalization. The data represent the mean ± SEM from three independent experiments.

**Figure 6 vetsci-09-00404-f006:**
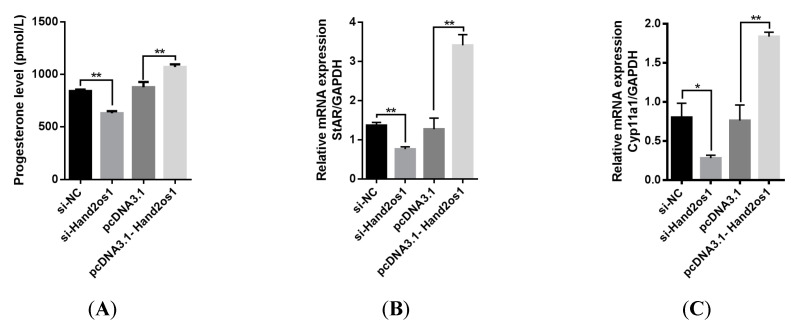
Effects of *Hand2os1* on progesterone production in LCs. LCs were isolated from the CL on D4 of pregnancy following transfection with si-*Hand2os1* or pcDNA3.1-*Hand2os1* for 24 h. Then, (**A**) the concentration of progesterone in the culture supernatants was determined by ELISA, and the expression levels of the steroidogenic enzyme (**B**) *StAR* and (**C**) *Cyp11a1* genes were analyzed by qRT-PCR. *GAPDH* was used as the reference gene for normalization. The data represent the mean ± SEM from three independent experiments. * *p* < 0.05, ** *p* < 0.01 compared with si-NC group, and different number of asterisks on bars indicate statistically significant differences (*p* < 0.05).

**Figure 7 vetsci-09-00404-f007:**
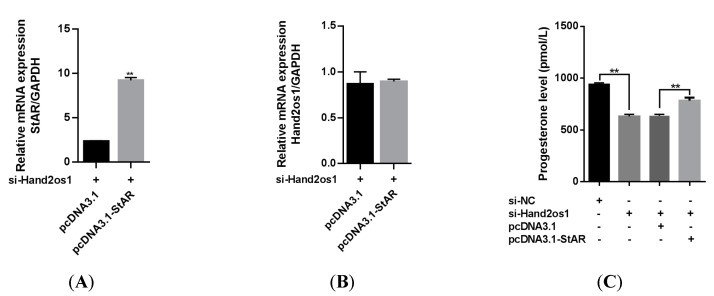
Effects of *StAR* overexpression on the progesterone production inhibited by si-*Hand2os1* in LCs. qRT−PCR analyses of (**A**) *StAR* and (**B**) *Hand2os1* mRNA levels 24 h after transfection with si-*Hand2os1* and pcDNA3.1-*StAR*. (**C**) The concentration of progesterone in the culture supernatants were determined by ELISA. *GAPDH* was used as the reference gene for normalization. The data represent the mean ± SEM from three independent experiments. ** *p* < 0.01 compared with si−NC group, and different number of asterisks on bars indicate statistically significant differences (*p* < 0.05).

**Table 1 vetsci-09-00404-t001:** Primer sequence for RT-qPCR.

Gene Name/Transcript ID	Gene Name/Transcript ID	Product Size (bp)
*Hand2*	F-GAGAACCCCTACTTCCACGG R-GACAGGGCCATACTGTAGTCG	71
*GAPDH*	F-TCACTGCCACCCAGAAGA R-GACGGACACATTGGGGGTAG	185
*Handos1*	F-GACAGAGTTGGAGATGGGCT R-GCAAGCACTTTCTCCCACTC	249
*StAR*	F-TAAACTCACTTGGCTGCTCAGTATTG R-GGTGGTTGGCGAACTCTATCTG	101
*Cyp11a1*	F-AGGTCCTTCAATGAGATCCCTT R-TCCCTGTAAATGGGGCCATAC	137
*Cyp19a1*	F-TGTGTTGACCCTCATGAGACA R-CTTGACGGATCGTTCATACTTTC	190

## Data Availability

Not applicable.

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
