# Peer review of "Hand2os1 Regulates the Secretion of Progesterone in Mice Corpus Luteum"

_vetsci, 2022, doi:10.3390/vetsci9080404_

Round 1
Reviewer 1 Report
Abstract
1. Line31: The ‘si-Hand2os1’ doesn’t make sense, please clarify.
2. No hypothesis or specific aim is stated; hence, the aim/justification and anticipated outcomes of the study remains elusive.
Figures
1. Figures 1 and 2: Cannot verify the validity of the results:
a. The figures are in disarray i.e., pictures are all over the text, the labels are separated from the text.
b. The pictures are very difficult to observe. Blue background completely overpowers/saturates any potential immunohistological detection. The pictures are too small. The magnification is too low, it needs to be increased to at least 40X.
c. Titles are not descriptive of the pictures, i.e., acronyms, times etc.
Discussion
1. Line 411-413: The conclusion study is overreaching. The authors did not test or determine that Hand2os1 is involved in the maintenance of CL function. They only demonstrated that it can participate in the regulation of steroidogenic process. Furthermore, the authors never stated the hypothesis so it is still unclear the overall intent of the study.
Animal use approval
1. Evidence for animal use approval in the ‘2.1 Ethics Statement’ was not provided, i.e., committee approval numbers for reference.
Note: Experimental controls do not appear to be a problem, which can be better determined upon revised immunohistochemical data. Nevertheless the manuscript require major revisions for reconsideration.
Reviewer 2 Report
The manuscript "Hand2os1 regulates the secretion of progesterone in mice corpus luteum" is interesting, well designed, and well written. Only a few minor mistakes need to be addressed before it is ready for publishing.
1)L112, 120, 128: reads "ovarian tissue from both sides of the body were collected", I would recommend that it is changed to "tissue from both ovaries was collected" or something similar.
2)Figure 1 has some issues, the labels inside the images are misplaced, please correct them.
3) Figures 3-7 have the legend "different number of asterisks on bars indicate statistical differences" but some figures have no asterisks, others have letters in the bars. Please clarify according to each figure.
Round 2
Reviewer 1 Report
Reviewer appreciates the revision efforts of the author. Minor editorial revisions are still needed to adequately guide the reader through the histology results. Additionally, the supplementary file was the same as the manuscript. No supplementary file was attached in the revised submission.
1. All figure titles should include definitions for acronyms/abbreviations used in the figure panels.
2. Magnifications corresponding to each needs to be included in the figure title text.
3. Day and hr should also be defined in the figure titles.
4. In figure titles where brackets are not utilized to indicate mean comparisons, the significant differences in the figure title text need to indicate the comparisons, i.e., means significantly different from controls or differences among all treatments.
